# Activation of Immune and Antiviral Effects by *Euglena* Extracts: A Review

**DOI:** 10.3390/foods12244438

**Published:** 2023-12-11

**Authors:** Yuji Isegawa

**Affiliations:** Department of Applied Biological Chemistry, Graduate School of Agriculture, Osaka Metropolitan University, Sakai, Osaka 599-8531, Japan; q22789t@omu.ac.jp

**Keywords:** *Euglena*, immunostimulation, antiviral activity, β-1,3-glucan, paramylon, minerals, influenza virus

## Abstract

Influenza is an acute respiratory illness caused by influenza virus infection, which is managed using vaccines and antiviral drugs. Recently, the antiviral effects of plants and foods have gained attention. *Euglena* is a motile unicellular alga and eukaryotic photosynthetic microorganism. It has secondary chloroplasts and is a mixotroph able to feed by photosynthesis or phagocytosis. This review summarizes the influenza treatment effects of *Euglena* from the perspective of a functional food that is attracting attention. While it has been reported that *Euglena* contributes to suppressing blood sugar levels and ameliorates symptoms caused by stress by acting on the autonomic nervous system, the immunostimulatory and antiviral activities of *Euglena* have also been reported. In this review, I focused on the immunostimulation of antiviral activity via the intestinal environment and the suppression of viral replication in infected cells. The functions of specific components of *Euglena*, which also serves as the source of a wide range of nutrients such as vitamins, minerals, amino acids, unsaturated fatty acids, and β-1,3-glucan (paramylon), are also reviewed. *Euglena* has animal and plant properties and natural compounds with a wide range of functions, providing crucial information for improved antiviral strategies.

## 1. Introduction

Sudden outbreaks of viruses, such as the recent outbreak of SARS-CoV-2, have occurred regularly. Particularly, influenza, a respiratory tract infection caused by influenza viruses, is a disease far more serious than the common cold syndrome. Although diverse vaccines and anti-influenza drugs are clinically used yearly, adapting the annual vaccines to the circulating influenza viruses prevalent during the influenza season is challenging. In addition, with the emergence of drug-resistant viruses, new approaches to treat influenza must be considered. Moreover, the risk of spread of infection through droplet and airborne transmission increases during disasters, especially in evacuation centers. Under these circumstances, it is reasonable that daily food ingredients can be incorporated into our daily lives to improve our physical condition, prevent diseases, and alleviate certain symptoms.

Furthermore, carbohydrates, proteins, and fats consumed daily are digested to produce cellular energy. Vitamins and minerals are indispensable coenzymes for this process; however, it is challenging for most people to consume sufficient amounts of these vitamins and minerals in their diets. To bridge the gap between a healthy lifespan and life expectancy, it is crucial to treat symptoms and disorders symptomatically while maintaining homeostasis in the body and mind to prevent their occurrence.

According to the Dietary Intake Standards for Japanese (2020 edition) [1], the target daily intake of dietary fiber is 21 g or more for men aged 18 and older (20 g or more for those aged 65 and older) and 18 g or more for women (17 g or more for those aged 65 and older). However, the dietary fiber intake is below the target for men and women across all age groups [1]. One source of dietary fiber that is attracting attention is β-glucan, a non-starch polysaccharide composed of D-glucose linked by beta-glycosidic bonds. The cell walls of various organisms are made of β-glucan. Therefore, we unintentionally consume β-glucan daily. Typical organisms from which β-glucan is derived include cereals (especially oats and barley), mushrooms, seaweeds, and yeast, with the structure differing depending on the source. In addition, β-glucan is not only a dietary fiber supplement but also has various functions. The functions of β-glucan have been recognized for more than half a century, and the effects of its intake on the human body include immunostimulatory [2,3,4,5], antiviral [6], regulation of intestinal microflora [7], lowering of cholesterol levels [2,8,9], prevention of diabetes [10], antioxidant [2,8,11,12,13], and anti-tumor [2,3,14,15,16,17,18]. The differences in the physiological functions of β-glucan may depend on its structure, including the composition of the glucan backbone, type and frequency of side chains, molecular size, solubility, and stereo structure.

*Euglena* is a microalga of approximately 0.05 mm in length and is characterized by its chlorophyll, photosynthesis, and use of flagella to perform cell deformation movements, called swimming and *Euglena* movements. It has secondary chloroplasts and is a mixotroph able to feed by photosynthesis or phagocytosis. Because of its plant and animal characteristics, it contains a wide range of nutrients, such as vitamins, minerals, amino acids, and unsaturated fatty acids (Table 1) [19,20,21,22]. The main storage product is paramylon, a β-1,3 polymer of glucose stored in the form of granules in the cytoplasm [23]. *Euglena* is a source of β-glucan, which has gained attention recently. It also contains nutrients such as paramylon [19,21,24], GABA (gamma-aminobutyric acid) [24], chlorophyll [25], lutein [26], zeaxanthin [26], spermidine [27], and putrescine [27]. *Euglena* utilizes a unique β-1,3-glucan called paramylon as a storage polysaccharide, making it one of the main components of *Euglena* as a food since it represents 30 to 80% of its dry weight, depending on the environment and culture method. As mentioned above, there are various types of β-glucans; however, paramylon is characterized by its linear and non-branched β-1,3-glucan structure [28,29], high crystal structure with a triple-helical structure [30,31], unique sugar composition (glucose only) [32], and insolubility in water. *Euglena* is also attracting attention as a functional food due to its nutritional value (Table 1).

The intake of *Euglena* may contribute to the suppression of high blood glucose levels [33] and the alleviation of stress-related symptoms [34] via autonomic nervous system balance; nonetheless, the effect of *Euglena* on the immunological and antiviral activities has not been studied in detail. Moreover, the active ingredients of *Euglena*, which contains a wide range of nutrients, have not been thoroughly analyzed. Unlike β-glucans from other sources, *Euglena* also serves as a broad source of vitamins, minerals, amino acids, unsaturated fatty acids, and other nutrients, and thus may exhibit a wide range of mechanisms of action. This review focuses on the immunostimulatory and antiviral activities of *Euglena*, with particular emphasis on its potential role in the intestinal environment. Furthermore, this review describes the effects of consuming *Euglena*, with a specific focus on its components.

## 2. Effects of *Euglena* Intake on Allergic Diseases

Allergic rhinitis (AR) is the most common allergic disease and represents a health problem worldwide. In Japan, the number of patients with AR has increased. Recently, the number of patients with pollinosis, particularly Japanese cedar pollinosis (JCP), has markedly increased beyond that associated with house dust mites (HDM) or pollinosis other than JCP. An epidemiological study revealed a marked increase (approximately 10%) in the prevalence of AR between 1998 and 2008 [35]. JCP is an immediate-type (type I) allergic disease that causes allergic symptoms due to a specific reaction with IgE antibodies. Cry j1 and Cry j2 in the pollen have been identified as the major antigens (allergens) causing JCP [36,37]. Although immunotherapy and drug therapy are available as treatments for JCP, there is still no effective cure. Recently, there have been reports that functional foods suppress allergic symptoms [38,39,40], raising the expectations for functional foods.

Helper T cells (Th), which are classified into Th1 and Th2 according to their cytokine production patterns, are involved in the pathogenesis of allergy [41]. Th1 participates in cellular immunity and secretes cytokines such as IL-2 and IFN-γ. Th2 is involved in humoral immunity and secretes cytokines such as IL-4 and IL-5. When Th1 and Th2 are unbalanced, allergies are thought to be triggered. For example, Th1 predominates in delayed (type IV) allergic development and Th2 in type I allergic development. Therefore, pollinosis is thought to develop when Th2 predominates.

*Euglena* is attracting attention as a new functional food, and the paramylon (β-1,3-D-glucan) contained in *Euglena* has immunostimulatory effects involving cytokines [42], hepatoprotective effects on acute liver injury [43] and anti-human immunodeficiency virus (HIV) [44], and antibacterial effects [45]. Furthermore, Sugiyama et al. [46] indicated that paramylon treatment could provide an effective alternative therapy for the management of atopic dermatitis (AD). Oral administration of paramylon was suggested to suppress the development of atopic dermatitis in NC/Nga mice, which spontaneously develop an eczematous AD-like skin lesion when kept under conventional care but not under specific pathogen-free (SPF) conditions [47,48], by inhibiting Th1 and Th2 responses [46]. The effects of *Euglena* and paramylon were observed in the early stages of A4gnt KO mice, which are mutant models that spontaneously develop gastric cancer through hyperplasia–dysplasia–adenocarcinoma mechanisms in the antrum of the stomach. The results suggest that the administration of *Euglena* and paramylon may ameliorate the early involvement of A4gnt mice through inflammatory modulation in the gastric mucosa [49]. Amorphous paramylon had a greater effect on intestinal immunity than paramylon, inhibiting colon cancer [50].

The production of allergen-specific IgE was significantly suppressed, and the production of IL-12 and IFN-γ increased when low-molecular-weight β-glucan was administered to mice [51]. It was also found that low-molecular-weight lentinan suppressed allergic symptoms, such as seasonal and perennial rhinorrhea, sneezing, nasal obstruction, itching, tearing in humans, and allergen-specific IgE and total IgE levels [52], indicating that the β-1,3-1,6-D-glucan found in mushrooms and yeast has immunomodulatory and allergy-suppressing effects. Although a decrease in serum IgE concentration was reported in atopic dermatitis-induced mice after the oral administration of paramylon, no decrease in IgE was observed in a pollinosis model mouse created by inoculation with Cry j1, because the hypersensitivity response to externally introduced specific antigens is biased toward Th2 dominance [53]. Furthermore, *Euglena* intake may directly reduce pollinosis symptoms, suggesting that components other than paramylon also relieve pollinosis [53].

## 3. Effects of *Euglena*’s Intake on the Intestinal Microbiota and Defecation

The intestinal tract hosts the gut microbiota, a complex bacterial community. The gut microbiota interacts with the host and strongly influences homeostasis and immunity in the host. Therefore, the gut microbiota is essential for maintaining the health of the host [54,55,56]. There is growing interest in optimizing the composition of the gut microbiota through dietary therapy using functional foods containing probiotics [57] or prebiotics [58].

For example, β-glucans in cereals are fermented by microorganisms living in the large intestine, and they are converted into short-chain fatty acids (SCFAs), such as acetic, propionic, and butyric acids [7]. SCFAs produced in the colon exert various effects, including immunomodulation [59], mediating apoptosis of colon cancer cells [60], and preventing obesity [61]. For example, the prevention of obesity is achieved by regulating energy metabolism via the SCFA receptor GPR43 and preventing the accumulation of excess lipids in adipose tissue [61]. In addition, SCFAs inhibit the growth of harmful bacteria such as Clostridium spp. and pathogenic Escherichia coli, thereby maintaining a healthy intestinal microflora [62].

The modulatory effects of diet on the gut microbiota are often investigated through in vivo studies in humans [63]. Information on the composition of the colon microbiota comes primarily from the analysis of fecal samples in human dietary intervention studies. However, this method has the experimental limitation that the production of certain metabolites, such as short-chain fatty acids (SCFAs), cannot be measured in situ (in the intestinal tract). A model culture system was developed to rigorously reproduce the microbial components of human fecal collections in vitro [64]. This in vitro human colon microbiota model was used to detect the decreased butyrate production in patients with ulcerative colitis [64]. Thus, combining in vitro human colon microbiota models with in vivo studies can help interpret changes in the human gut microbiota.

Since the effects of *Euglena*’s ingestion on the human gut microbiota are not yet clear, this section evaluates the effects of *Euglena* on the colon microbiota of healthy human subjects. Furthermore, by analyzing the effects of adding *Euglena* or paramylon to an in vitro human colon microbiota model, verifying the effects of *Euglena* on the gut microbiota is feasible.

Studies on the effects of *Euglena*’s intake on gut microbiota and defecation showed that the occupancy of the genus *Faecalibacterium* was increased by *Euglena* in vitro and in vivo [65]. However, this effect may not be due to the paramylon contained in *Euglena* [65]. *Faecalibacterium prausnitzii* is one of the butyrate-producing bacteria with the highest occupancy in the intestinal tract [66]. Other butyrate-acid-producing bacteria, such as *Roseburia*, showed no differences. Thus, the changes in the gut microbiota induced by *Euglena* may be specific to *Faecalibacterium*. Gao et al. reported that butyrate improves insulin resistance [67] and Jia et al. showed that increasing the number of butyrate-producing bacteria may be useful to treat type 2 diabetes [68]. Such results of increased butyrate production induced by *Euglena* are discussed with the results of a study showing that *Euglena* consumption lowered blood glucose levels in a rat model of type 2 diabetes [33], indicating that butyrate production by the gut microbiota may be one of the mechanisms by which blood glucose levels are reduced. *Euglena* is also a source of vitamins, minerals, and unsaturated fatty acids [69,70]. These components promote the growth of *Faecalibacterium*. Okouchi et al. showed that ingestion of *Euglena* increased bifidobacteria in the intestinal microflora of mice [71]. However, in another experiment, there was no significant increase in the relative occupancy of bifidobacteria in vitro or in vivo. Therefore, *Euglena* consumption may promote acetic acid production by *Bifidobacterium*, which is then consumed by *Faecalibacterium*, promoting *Bifidobacterium* growth and butyrate production. In *Bifidobacterium adolescentis* and *F. prausnitzii*, a cross-feeding process has been reported using a carbon source of fructooligosaccharides [72].

Furthermore, stool volume was increased by the ingestion of *Euglena* [65], as shown by Asayama et al. [73]. These results are consistent with the findings of Kawano et al. that rats fed a diet containing cholesterol and *Euglena* had a shorter cholesterol retention time in the intestine than rats fed a diet containing only cholesterol [74]. Dietary fiber intake may increase stool frequency in patients with constipation [75]. The paramylon in *Euglena* is insoluble and is neither digested nor absorbed. Therefore, it is believed to exert the same effect as a dietary fiber. Moreover, the consumption of *Euglena* enhanced butyrate production by *F. prausnitzii* and is consistent with the results of a previous study showing that butyrate supplementation may alleviate defecation disorders [76]. Thus, ingesting *Euglena* may have beneficial effects on constipation, such as reducing pain during defecation. In addition, butyrate produced in the intestine increases the number of specific CD8+ T cells that eliminate influenza viruses and control the infection [77]. These findings indicate *Euglena* may contribute to immunomodulation and antiviral activity (Figure 1).

Overall, these findings suggest that *Euglena* increases the occupancy of *Faecalibacterium*, which in turn promotes butyrate production (Figure 1). Future challenges for these studies include increasing the sample size for the in vivo analysis and identifying active components other than paramylon in *Euglena*. Thus, *Euglena* has great potential as a novel prebiotic.

## 4. Effect of *Euglena* Intake on Symptoms of Influenza Virus Infection

Influenza is an acute respiratory illness caused by the influenza virus [78,79]. Influenza increases in winter and is a serious social and economic problem in many countries. Influenza virus infection causes multiple systemic symptoms, including fever over 38 °C, headache, arthralgia, myalgia, and fatigue. Typically, healthy adults recover spontaneously without antiviral treatment through self-healing mechanisms involving the immune response. However, in the elderly, infants, and patients with an underlying respiratory disease, impaired immune function can exacerbate symptoms due to viral infection and, in the worst case, pneumonia or encephalitis, which can be fatal. Because influenza viruses are segmental-stranded RNA viruses, gene replication and reassortment are frequent. Therefore, vaccination does not provide adequate protection against influenza virus infection.

In the previous section, it was shown that *Euglena* consumption contributes to the regulation of butyrate-producing bacteria occupancy in the gut and may contribute to antiviral activity by stimulating butyrate production; however, the effect of *Euglena* or paramylon on influenza virus infection is unknown.

This section reviews the alleviating effects of *Euglena* and paramylon on influenza virus infection symptoms in mice based on survival, lung virus titer, and cytokine production.

*Euglena* or paramylon administration prevented a decrease in survival after influenza virus infection. High lung virus titers and/or abnormal production of inflammatory cytokines frequently occur with the progression of influenza virus infection and are associated with the severity of the morbidity [80,81,82,83]. Therefore, lung homogenates from infected mice were used to measure lung viral titers and cytokine production. On day 1 post-infection, pulmonary viral titers were similar in the *Euglena-* and paramylon-fed groups. However, the paramylon-treated group showed lower viral titers on day 2 compared to those of the *Euglena*-treated group. The production of IL-1β, IFN-β, IFN-γ, and TNF-α was higher in the *Euglena* group than in the control group. Significant increases in IL-1β, IL-6, IL-12, IL-10, IFN-γ, and TNF-α were detected in the paramylon-treated group, and increased production of IFN-β was observed. These results suggest that paramylon is one of the functional substances in *Euglena* that alleviate influenza virus infection symptoms. The *Euglena* used in this experiment contained approximately 30% paramylon of a β-1,3-glucan.

Recent immunological studies have shown that influenza virus infection induces the production of type I interferons such as INF-β, leading to acute inflammation of the lungs, after which INF-β contributes to viral elimination via induction of NK and CD8+ T cells [84,85]. It has also been reported that β-1,3-glucan enhances NK cell activity in influenza virus-infected mice, increases IL-1β, TNF-α, and IFN-γ levels in the lungs [6,86], activates type I interferon, and induces CD8+ T cells via Dectin-1 in dendritic cells [87]. Furthermore, the proliferation, deviation, and activation of NK cells are induced by the IFN, TNF-α, and IL-1 secreted by dendritic cells and macrophages [88,89,90].

Generally, Dectin-1, a C-type lectin-like pattern recognition receptor on the surface of leukocytes, is the primary receptor for β-glucan [91]. The triple-helix structure formed by the backbone of β-1,3-glucan is specifically recognized by Dectin-1. As mentioned above, the crystal structure of paramylon consists of a triple helix [30,31], which is recognized by the β-glucan receptor Dectin-1, and it is presumed that paramylon activates Dectin-1. Moreover, the size of paramylon is typically 2–3 μm, which is comparable to the size of pathogenic bacteria. This suggests paramylons may pass through epithelial cells using the same mechanism as pathogenic bacteria. Indeed, consistent with the above data, paramylon has been shown to bind directly to the recombinant Dectin-1 [92] and upregulate inflammatory factors such as NO, TNF-α, IL-6, and COX-2 [93]. Studies have shown that the number and size of β-1,6 branches on the β-1,3 skeleton are crucial for the function of β-glucans [94,95]; however, despite its limited number of branches, paramylon shows functional diversity. In the results of the study by Nakashima et al. [96], mice were fed the same amount of *Euglena* and paramylon; however, the survival rate and viral titer in the lungs results tended to be better in the paramylon-fed than in the *Euglena*-fed group. However, the *Euglena*-fed group did not show the same results as those of the paramylon-fed group. In particular, the *Euglena*-fed group showed a different pattern of cytokine production, suggesting the involvement of components other than paramylon in *Euglena* treatment. These results indicate that *Euglena* may prevent influenza virus infection not only through the actions of components other than paramylon in the intestinal microflora, but also through systemic immune regulation, mainly through the contribution of paramylon in *Euglena*.

Oral intake of *Euglena* and paramylon significantly reduced pulmonary viral titers and increased survival. In addition, paramylon induced significant increases in cytokine levels in the lungs (Figure 2). However, the pattern of cytokine production in the *Euglena*-fed group did not completely match that of the paramylon-fed group, suggesting the involvement of components other than the paramylon contained in *Euglena*. The oral intake of *Euglena* and paramylon can eliminate influenza viruses, mainly through the activity of β-1,3-glucan on dendritic cells and induction of CD8+ T and/or NK cells (Figure 2). Further research is required to gain a more comprehensive understanding of *Euglena* and paramylon’s functions and their potential for influenza prevention, possibly through their direct effects on the viral replication cycle.

## 5. Effect of *Euglena* on Cellular Infection with the Influenza Virus

Influenza is an infectious respiratory tract disease caused by A, B, or C influenza viruses. Type A and C are the most and least common, respectively. Influenza A and B viruses, the most prevalent types, comprise two surface glycoproteins, hemagglutinin (HA) and neuraminidase (NA), which are antigens against the host targets for protective immunity. The antigenic properties of these viruses differ depending on the combination of HA and NA subtypes. Viruses with various combinations of these glycoproteins exist in humans and other parts of the animal kingdom. Accumulation of mutations in HA and NA genes gradually changes the antigenicity of the virus, and new strains emerge, even of the same subtype. Influenza viruses are endemic every year because of frequent antigenic mutations. Generally, after an incubation period of 1–3 days after influenza virus infection, symptoms such as fever (usually higher than 38 °C), headache, general malaise, and muscle and joint pain suddenly occur. This is followed by upper respiratory tract inflammatory symptoms, such as cough and nasal discharge, and symptoms usually abate after one week. Patients, particularly the elderly, with chronic diseases of the respiratory, cardiovascular, and renal systems, or with metabolic diseases such as diabetes, and patients of any age with compromised immune function are more susceptible to secondary bacterial infections of the respiratory system, which can worsen the original disease and increase the risk of hospitalization or death. In children, influenza can also cause otitis media, febrile convulsions, and bronchial asthma. Currently, vaccines are used to prevent influenza virus infection, and antiviral drugs are used for treatment. However, there are concerns about the side effects of adjuvants contained in existing influenza vaccines, the inability to respond rapidly to new viruses, and the fact that these vaccines are less effective in prevention than vaccines against other viruses. Furthermore, the side effects of antiviral drugs are equally problematic, as is the emergence of drug-resistant bacteria [98,99,100,101]. Thus, there is a need to develop new preventive and therapeutic methods to overcome these issues. In light of this, we have focused on the functional properties of foods and investigated the mechanisms of action of components with antiviral activity. The previous section mentioned that mice fed a diet containing *Euglena* powder and subsequently infected with the influenza virus showed improved survival [96]. Based on the pattern of cytokine production in these mice, it was speculated that paramylon, which is mainly contained in *Euglena*, may have contributed to the elimination of the virus by mobilizing the systemic immune system. However, the antiviral effect of *Euglena* may have mechanisms other than systemic immunity. In fact, the cytokine production pattern of the *Euglena*-fed group in the in vivo experiment described in the previous section did not completely match that of the paramylon-fed group, suggesting that components other than the paramylon contained in *Euglena* may be involved. The direct mechanism by which other components of *Euglena* suppress viruses in cells was reported by Nakajima et al. [102]. In vitro experiments are necessary to examine the direct effect; however, the paramylon contained in *Euglena* is an insoluble β-glucan that is difficult to add to the culture medium, making it difficult to conduct in vitro experiments. Therefore, this section aims to clarify the antiproliferative effect of *Euglena* hot water extract on the influenza virus and its mechanism.

It was confirmed that *Euglena* hydrothermal extract suppresses virus proliferation using MDCK cells infected with various influenza virus strains. In particular, the results showed that the *Euglena* hot water extract was effective against oseltamivir-resistant virus strains, suggesting that the mechanism of inhibition of influenza virus growth by the extract is different from that of oseltamivir [102]. In addition, no significant difference in IC_50_ values was observed among the virus strains examined. Therefore, the virus replication inhibitory activity of the *Euglena* hot water extract does not show virus specificity [102]. This differs from amantadine, which is effective against type A influenza virus strains but ineffective against type B strains [103].

Once the virus attaches to the cell membrane, it is endocytosed, and RNA is released from the viral particles into the cytoplasm, where it is transferred to the nucleus for replication and transcription, followed by the synthesis of viral proteins and the viral genome. Once the virus components are in place, the budding phase of the viral growth cycle occurs, when the viral particles aggregate near the cell membrane and are released via neuraminidase activity. One growth cycle lasts approximately eight hours, and studying the inhibition of the viral process is possible. "Relenza" and "Tamiflu," which are the mainstream anti-influenza drugs, inhibit the budding phase of the virus by blocking neuraminidase activity [104], whereas the recently launched "Xofluza" inhibits viral RNA replication [105]. Although the mechanism of inhibition by *Euglena* hydrothermal extract has not yet been predicted, *Euglena* hydrothermal extract reduced the viral titer by pretreatment or prolonged treatment of infected cells [102]. These results suggest that the *Euglena* hydrothermal extract activates host cell defense mechanisms.

In the mouse infection experiments described in the previous section, oral ingestion of *Euglena* alleviated the symptoms of influenza virus infection, mainly through the contribution of paramylon. The ingestion of paramylon by mice reduced the amount of interferon β in their blood, which the virus infection had increased, on day 3 [96]. After insoluble paramylon is recognized by Dectin-1 (a major β-glucan receptor expressed on intestinal immune cells, such as dendritic cells and macrophages) and endocytosed, the activation of tyrosine-protein kinase SYK and transcription factor nuclear factor-kappa B promote cytokine secretion [92,97]. The hydrothermal extract of *Euglena* consists of water-soluble components and low-molecular-weight substances, such as polyphenols, from which insoluble paramylons are thought to be excluded. The concentration of carbohydrates in the hydrothermal extract of *Euglena* was 0.53% using the phenol sulfate method, whereas the concentration of carbohydrates in the *Euglena* powder was approximately 30–40%. However, to discard the possibility that a small amount of glucan was involved, the hydrothermal extract of *Euglena* was treated with β-glucanase to verify if it affected its antiviral activity, and it was found to have no effect. This suggests that β-glucan is not involved in the in vitro influenza virus growth suppression observed with the hydrothermal extract of *Euglena*.

Recently, it has been confirmed that the hydrothermal extract of *Euglena* suppresses lung cancer symptoms in mice by stimulating host immunity [106]. Therefore, components other than paramylon may be involved in such immune mechanisms. Generally, polyphenols exhibit antiviral effects [107]. In the case of the hydrothermal extract of *Euglena*, as β-glucan and polyphenols showed no antiviral activity, the minerals were analyzed [102]. Zinc is required for the growth of *Euglena* and accumulates in the cells. In addition, compared to other metals, zinc has antiviral activity by promoting the induction of type 1 interferon receptors, thereby inducing the production of the antiviral protein 2’–5’ oligoadenylate synthetases [108]. In the case of SARS-CoV and equine arteritis virus (EAV), the RNA synthesis is catalyzed by an RNA-dependent RNA polymerase, which is directly inhibited by zinc [109]. Specifically, zinc blocks the initiation step of EAV RNA synthesis, whereas the RNA-dependent RNA polymerase elongation is inhibited and template binding reduced in the case of the SARS-CoV [109]. Nakashima et al. [102] showed that the addition of zinc acetate inhibited influenza virus growth (Figure 3). Furthermore, the addition of zinc acetate to the demineralized *Euglena* extract restored its influenza virus inhibitory activity. The CER (cation exchange resin)-treated *Euglena* extract, which has an equivalent zinc concentration, has higher anti-influenza virus inhibitory activity than zinc alone [102]. Ionophores are required for zinc uptake into cells [109]. *Euglena* may contain substances that act similarly to ionophores, whereas the hydrothermal extract of *Euglena* may contain higher concentrations. The ionophores may capture zinc, ensuring some anti-influenza virus activity of the membrane-treated extract at high concentrations. Therefore, the effect of zinc acetate was retained in the normal *Euglena* extract because of the presence of free zinc, but in CER-treated *Euglena*, the removal of free zinc by membrane treatment allowed the ionophores to exert the effect of the zinc captured by the ionophores. This suggests that substances other than zinc are also involved in the anti-influenza virus activity of *Euglena*.

This indicates that virus elimination is not only caused by the paramylon in *Euglena* eliminating viruses by activating the systemic immunity, but also by the action of other components of *Euglena*, including zinc, which may act directly on cellular defense mechanisms. A limitation is that the MDCK cells used in the aforementioned study are regularly used in viral research, whereas cells of other origins should also be investigated.

In conclusion, hydrothermal extracts of *Euglena* reduced the viral titer of various influenza viruses in vitro, and its inhibitory activity was more potent when host cells were pre-exposed to hydrothermal extracts of *Euglena*, suggesting that *Euglena* components may provide signals to enhance host cell defense mechanisms. Furthermore, the *Euglena* extract inhibits the growth of influenza viruses by a nonspecific mechanism different from those of existing drugs. Therefore, this extract may be a promising treatment for infections caused by newly mutated influenza virus strains.

## 6. Conclusions

In this review, I focused on the immunostimulation of antiviral activities and its potential contribution to the intestinal environment. While reviewing whether the diverse components of *Euglena* are involved in the antiviral activity, the beneficial effects of *Euglena*’s intake became clear.

In Section 2, *Euglena* extract and its components, such as paramylon, were shown to be effective in alleviating allergies such as hay fever. In Section 3, it was shown that *Euglena* components other than paramylon may affect *Faecalibacterium* occupancy and promote defecation by increasing butyrate production [65]. Previous studies have shown that butyrate production from the intestinal flora increases the number of specific CD8+ T cells that eliminate influenza viruses and control infection, and these findings indicate that *Euglena* may be involved in immunomodulation and antiviral activity. In anticipation of its involvement in immunity by improving the intestinal environment, the suppression of influenza virus infection symptoms by *Euglena*’s intake in vivo was confirmed [96]. Thus, influenza virus infection could be prevented not only through the action of the intestinal microflora mediated by *Euglena* components, but also through the systemic immune regulation mediated by the contribution of paramylon in *Euglena*. In Section 5, the direct action of *Euglena* on influenza virus infection was confirmed [102]. Since this review focused on in vitro experiments and assumed the involvement of water-soluble components in *Euglena*, studies with the hydrothermal extracts of *Euglena*, excluding the insoluble paramylon, were reviewed. These studies showed that the hydrothermal extract of *Euglena* strongly inhibited the infection of all influenza virus strains, including strains resistant to the anti-influenza drugs, indicating that the hydrothermal extract of *Euglena* may stimulate the host cell defense mechanism rather than acting directly on the replication machinery of the influenza virus. Furthermore, it became clear that the minerals contained in the hydrothermal extract of *Euglena*, especially zinc, were involved in the infection-inhibitory action of the extract. These studies indicate not only that the paramylon in *Euglena* eliminates viruses by systemic immunity, but also that other components of *Euglena*, including zinc, may exert antiviral activity by directly acting on cellular defense mechanisms (Figure 4).

Thus, *Euglena* is effective in the prevention and treatment of influenza through the improvement of intestinal microflora, as well as through the regulation of systemic immunity and the enhancement of cellular defense mechanisms. These functions were associated with one of the major components, paramylon, and the combined contribution of other components in *Euglena*. This suggests the value of eating whole *Euglena* as a source of β-glucan (Figure 4).

I propose the ingestion of *Euglena* as a strong candidate for daily health management through food, prevention against viral infections, and alleviation of symptoms of influenza.

## Figures and Tables

**Figure 1 foods-12-04438-f001:**
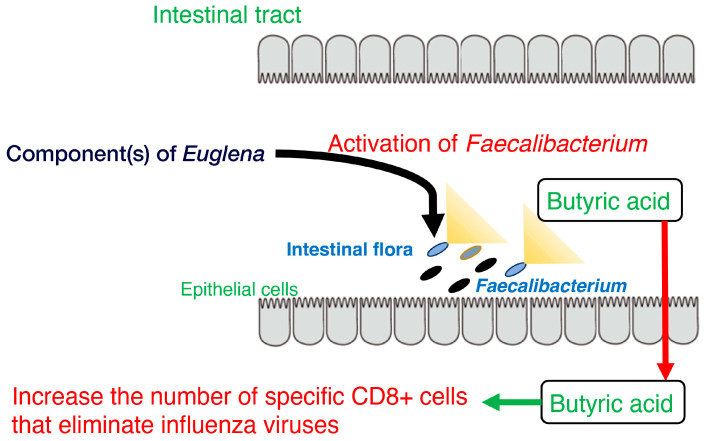
Model of the activation of *Faecalibacterium*, butyrate production, and CD8+ T cells by component(s) of *Euglena*.

**Figure 2 foods-12-04438-f002:**
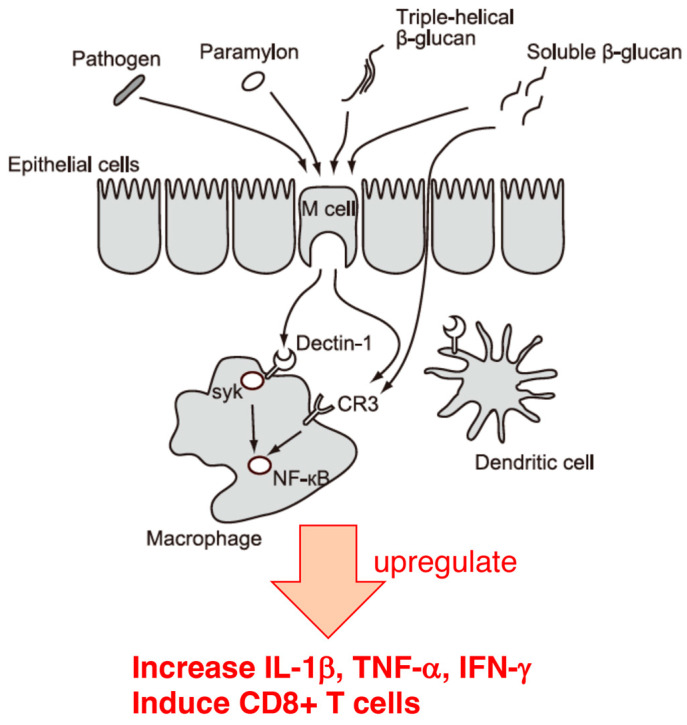
Immune regulation by paramylon. This is a diversion arrangement of Figure 3 from Nakajima et al. [97]. Reprinted with permission from Ref. [97]. 2023, Nakashima, A. et al.

**Figure 3 foods-12-04438-f003:**
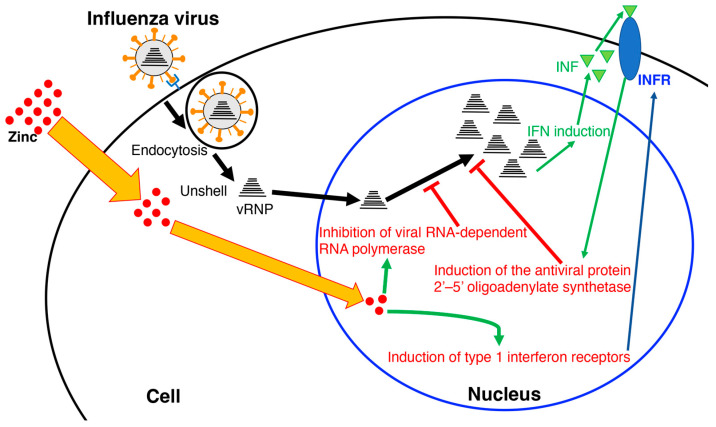
Model of the inhibition of influenza virus replication by zinc. INF: type 1 interferon; INFR: type 1 interferon receptor; vRNP: viral ribonucleoprotein.

**Figure 4 foods-12-04438-f004:**
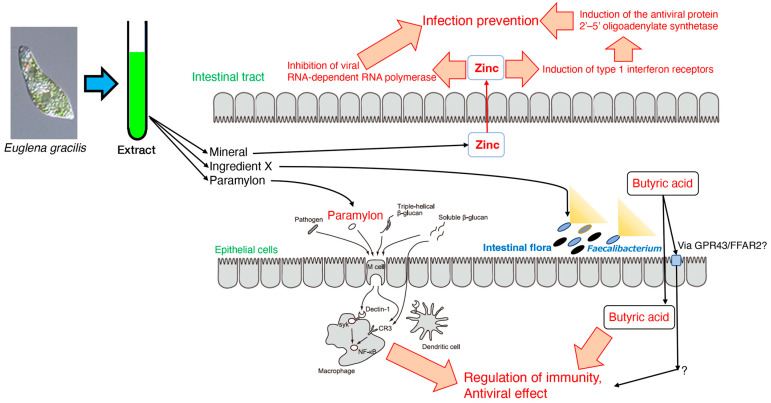
Model of the anti-influenza virus effect by *Euglena* extract.

**Table 1 foods-12-04438-t001:** Nutrients contained in *Euglena*.

Vitamins			Minerals			Amino Acids		Unsaturated Fatty Acids	
	Content	Ref.		Content	Ref.		Ref.		Ref.
	(µg/10^6^ cells)			(µg/10^6^ cells)					
Vitamin B_1_	1.4 ± 0.1	[19]	Zn^2+^	0.795 ± 0.104	[19]	Alanine	[20]	Docosahexaenoic acid	[21,22]
Vitamin B_2_	3.5 ± 0.2	[19]	Mg^2+^	0.165 ± 0.076	[19]	Arginine	[20]	Eicosapentaenoic acid	[21]
Vitamin B_6_	7.5 ± 0.6	[19]	Fe^2+^	0.153 ± 0.092	[19]	Aspertic acid	[20]	α-Linolenic acid	[19,21]
Vitamin B_12_	0.05 ± 0.01	[19]	Mn^2+^	0.240 ± 0.056	[19]	Cystine	[20]	Arachidonic acid	[19,21]
Folic acid	1.7 ± 0.5	[19]	Ca^2+^	0.046 ± 0.070	[19]	Glycine	[20]	Palmitoleic acid	[21]
Pantotheric acid	18.2 ± 0.4	[19]	Cu^2+^	0.004 ± 0.0023	[19]	Glutamic acid	[20]	Oleic acid	[19,21]
Biotin	4.6 ± 0.3	[19]	Ni^2+^	0.006 ± 0.0014	[19]	Histidine	[20]	Linoleic acid	[19,21]
Vitamin C	27.2 ± 0.4	[19]				Isoleucine	[19,20]	Eicosadienoic acid	[21]
Tocopherol	517.5 ± 10.2	[19]				Leucine	[19,20]	Docosatetraenoic Acid	[21]
β-Carotene	0.71 ± 0.15	[19]				Lysine	[19,20]	Docosapentaenoic acid	[21,22]
						Phenylalanine	[20]	Dihomo-γ-linolenic acid	[21]
						Preoline	[20]		
						Serine	[20]		
						Threonine	[19,20]		
						Tryptophan	[19,20]		
						Tyrosine	[20]		
						Varine	[20]		

## Data Availability

No new data were created or analyzed in this study.

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
