# Peer review of "Activation of Immune and Antiviral Effects by *Euglena* Extracts: A Review"

_foods, 2023, doi:10.3390/foods12244438_

Round 1
Reviewer 1 Report
Comments and Suggestions for Authors
Abstract:
1. The sentence "While it has been reported that Euglena contributes to suppressing blood sugar levels and ameliorates symptoms caused by stress by acting on the autonomic nervous system, the immunostimulatory and antiviral activities of Euglena have also been reported." is quite long and could be made clearer by breaking it into two sentences.
2. The term "Euglena" is mentioned throughout the abstract, but it might be beneficial to provide a brief explanation of what Euglena is for readers who may not be familiar with it.
Introduction
Overall, the introduction is well-structured and informative, but there are a few grammatical and scientific errors that should be addressed.
1. The sentence "To fundamentally reduce the gap between a healthy life span and expectancy, it is essential to treat the symptoms and disorders symptomatically and maintain homeostasis in the body and mind for them not to occur" is somewhat convoluted. It could be rewritten for better clarity, such as: "To bridge the gap between a healthy lifespan and life expectancy, it is crucial to treat symptoms and disorders symptomatically while maintaining homeostasis in the body and mind to prevent their occurrence."
2. In the sentence, "The differences in the physiological functions of β-glucan may depend on its structure, including the composition of the glucan backbone, type and frequency of side chains, molecular size, solubility, and steric structure," the word "steric" should be replaced with "stereo" for the correct terminology, as "steric" is related to spatial arrangement in chemistry.
3. In the sentence, "Euglena is a source of β-glucan that has gained attention recently," it could be beneficial to provide a brief explanation of what Euglena is for readers who may not be familiar with it.
4. The sentence "In this review, I focused on the immunostimulation of antiviral activity and the possible contribution from the intestinal environment, which is one of the mechanisms of immunostimulation" is clear but could be made more concise. For example, you could say, "This review focuses on the immunostimulatory and antiviral activities of Euglena, with a particular emphasis on its potential role in the intestinal environment."
5. In the sentence "Moreover, the effects of Euglena's intake, focusing on its components and effects, are also described," the use of "effects" twice in close proximity may be redundant. Consider rephrasing for clarity, such as, "Furthermore, this review describes the effects of consuming Euglena, with a specific focus on its components."
6. The sentence "We believe that further studies are needed to better understand the detailed functions of Euglena and paramylon and to enable influenza prevention" could be rephrased for clarity. For instance, "Further research is required to gain a more comprehensive understanding of Euglena and paramylon's functions and their potential for influenza prevention."
Comments on the Quality of English Language
Minor editing of the English language required
Author Response
Reviewer 1: Comments and Suggestions for Authors
Abstract:
- The sentence "While it has been reported that Euglena contributes to suppressing blood sugar levels and ameliorates symptoms caused by stress by acting on the autonomic nervous system, the immunostimulatory and antiviral activities of Euglena have also been reported." is quite long and could be made clearer by breaking it into two sentences.
Response: Thank you for your pertinent comment. Accordingly, I separated the sentence into two.
- The term "Euglena" is mentioned throughout the abstract, but it might be beneficial to provide a brief explanation of what Euglena is for readers who may not be familiar with it.
Response: Thank you for your pertinent comment. I added explanatory text for Euglena.
Introduction
Overall, the introduction is well-structured and informative, but there are a few grammatical and scientific errors that should be addressed.
- The sentence "To fundamentally reduce the gap between a healthy life span and expectancy, it is essential to treat the symptoms and disorders symptomatically and maintain homeostasis in the body and mind for them not to occur" is somewhat convoluted. It could be rewritten for better clarity, such as: "To bridge the gap between a healthy lifespan and life expectancy, it is crucial to treat symptoms and disorders symptomatically while maintaining homeostasis in the body and mind to prevent their occurrence."
Response: Thank you for your pertinent comment. I replaced it with your recommended sentence.
- In the sentence, "The differences in the physiological functions of β-glucan may depend on its structure, including the composition of the glucan backbone, type and frequency of side chains, molecular size, solubility, and steric structure," the word "steric" should be replaced with "stereo" for the correct terminology, as "steric" is related to spatial arrangement in chemistry.
Response: Thank you for your pertinent comment. I replaced it with “stereo”.
- In the sentence, "Euglena is a source of β-glucan that has gained attention recently," it could be beneficial to provide a brief explanation of what Euglena is for readers who may not be familiar with it.
Response: Thank you for your comment. I added further details on Euglena and paramylon based on your comment and added a reference for paramylon as a storage carbon source.
- The sentence "In this review, I focused on the immunostimulation of antiviral activity and the possible contribution from the intestinal environment, which is one of the mechanisms of immunostimulation" is clear but could be made more concise. For example, you could say, "This review focuses on the immunostimulatory and antiviral activities of Euglena, with a particular emphasis on its potential role in the intestinal environment."
Response: Thank you for your pertinent comment. I replaced it with the recommended sentence.
- In the sentence "Moreover, the effects of Euglena's intake, focusing on its components and effects, are also described," the use of "effects" twice in close proximity may be redundant. Consider rephrasing for clarity, such as, "Furthermore, this review describes the effects of consuming Euglena, with a specific focus on its components."
Response: Thank you for your pertinent comment. I replaced it with the recommended sentence.
- The sentence "We believe that further studies are needed to better understand the detailed functions of Euglena and paramylon and to enable influenza prevention" could be rephrased for clarity. For instance, "Further research is required to gain a more comprehensive understanding of Euglena and paramylon's functions and their potential for influenza prevention."
Response: Thank you for your pertinent comment. I replaced it with the recommended sentence.
Comments on the Quality of English Language: Minor editing of the English language required
Response: The service of Editage (www.editage.com) was sought for language editing of the revised version of the manuscript.
Reviewer 2 Report
Comments and Suggestions for Authors
Influenza is an acute respiratory illness caused by influenza virus infection, which is managed using vaccines and antiviral drugs. Recently, the antiviral effects of plants and foods have gained attention. This review summarizes the therapeutic properties of Euglena against influenza from a functional food viewpoint. Review, is focused on the immunostimulation of antiviral activity via the intestinal environment and the suppression of viral replication in infected cells.
The publication thus fits into the latest research areas and covers important topics for food and health care. It is written comprehensively and contains more important information related to the consumption of Euglena and its impact on allergenic diseases, on human intestinal microflora and its impact on the influenza virus. The information contained in the publication was additionally presented graphically, especially the course of reactions occurring at the cell level. The presented conclusions fully reflect the presented assumptions of the work. The cited literature is extensive and unobjectionable, and is adequate to the topic covered in the publication.
Author Response
Reviewer 2: Comments and Suggestions for Authors
Influenza is an acute respiratory illness caused by influenza virus infection, which is managed using vaccines and antiviral drugs. Recently, the antiviral effects of plants and foods have gained attention. This review summarizes the therapeutic properties of Euglena against influenza from a functional food viewpoint. Review, is focused on the immunostimulation of antiviral activity via the intestinal environment and the suppression of viral replication in infected cells.
The publication thus fits into the latest research areas and covers important topics for food and health care. It is written comprehensively and contains more important information related to the consumption of Euglena and its impact on allergenic diseases, on human intestinal microflora and its impact on the influenza virus. The information contained in the publication was additionally presented graphically, especially the course of reactions occurring at the cell level. The presented conclusions fully reflect the presented assumptions of the work. The cited literature is extensive and unobjectionable, and is adequate to the topic covered in the publication.
Response: Thank you for taking the time to critically evaluate our manuscript.
I sincerely appreciate your encouraging comments. I added details to the text based on the comments and added more references. Please review our revised manuscript.